# The Cytotoxicity of OptiBond Solo Plus and Its Effect on Sulfur Enzymes Expression in Human Fibroblast Cell Line Hs27

Anna Bentke-Imiolek [1], Kinga Kaszuba [1], Patrycja Bronowicka-Adamska [1], Barbara Czopik [2], Joanna Zarzecka [2] and Maria Wróbel [1,*]

1   Jagiellonian University Medical College, Faculty of Medicine, Chair of Medical Biochemistry, 31-034 Cracow, Poland; anna.bentke-imiolek@uj.edu.pl (A.B.-I.); kinga.kaszuba@uj.edu.pl (K.K.); patrycja.bronowicka-adamska@uj.edu.pl (P.B.-A.)
2   Jagiellonian University Medical College, Faculty of Medicine, Department of Conservative Dentistry with Endodontics, Institute of Dentistry, 31-155 Cracow, Poland; barbara.czopik@uj.edu.pl (B.C.); j.zarzecka@uj.edu.pl (J.Z.)
*   Correspondence: mtk.wrobel@uj.edu.pl

**Abstract:** The aim of the study was to determine the cytotoxic concentrations and incubation times of the commonly used dental adhesive system OptiBond Solo Plus in its non-polymerized form, and to test how it relates to oxidative stress by determining the reduced and oxidized glutathione (GSH and GSSG) levels as well as to study its influence on cell number and the expression of selected sulfur enzymes, with particular emphasis on cystathionine γ-lyase (CTH) and 3-mercaptopyruvate sulfurtransferase (MPST). All investigations were conducted on an in vitro model of human fibroblast cell line Hs27. Changes in cellular plasma membrane integrity were measured by the LDH test. The expression levels were determined by RT-PCR and Western blot protocols. Changes in cell number were visualized using crystal violet staining. The RP-HPLC method was used to determine the GSH and GSSG levels. Reduced cell number was shown for all tested concentrations and times. Changes in the expression on the mRNA and protein level were demonstrated for CTH and MPST enzymes upon exposure to the tested range of OptiBond concentrations. Levels of low-molecular sulfur compounds of reduced and oxidized glutathione were also established. Cytotoxic effect of OptiBond Solo Plus may be connected with the changes of MPST and CTH sulfur enzymes in the human fibroblast Hs27 cell line. The elevated levels of these enzymes could possibly show the antioxidant response to this dental adhesive system. OptiBond Solo Plus in vitro results should be taken into consideration for further in vivo tests.

**Keywords:** OptiBond Solo Plus; dental materials; dental adhesive; bonding system; fibroblasts; cystathionine γ-lyase; 3-mercaptopyruvate; sulfurtransferase

## 1. Introduction

Adhesive bonding systems are one of the most commonly applied dental materials in everyday clinical practice. They are used for bond restoration with dentin and enamel and the proper bonding strength can be achieved only when those systems are polymerized properly. In orthodontic treatment, the bonding process poses a threat of monomers coming into direct contact with the living tissue within the oral cavity [1–3]. Incomplete polymerization of dental adhesives can be a source of these monomers. Additionally, permeability of simplified adhesives corelates with incomplete polymerization of resin monomers. This problem also refers to OptiBond family dental adhesives [4,5]. Modern universal adhesive systems can be used in self-etch, etch-and-rinse, or a selective enamel-etching mode. Their bonding efficiency is still being investigated, as the durability and stability of the dentin–adhesive interface are limited [6]. Various studies have described techniques or strategies available to enhance universal adhesives and dentin bond strength including OptiBond Solo Plus [7]. Even so, adhesive systems lose their connection to dentin with time, and the

degradation is related to that loss of bond strength [3]. In other words, polymerized dental products break down over time and release un-polymerized components, exposing surrounding living tissues—especially the pulp—to potential noxious effects [8]. Traditionally used methacrylate chemicals described in in vitro studies indicate that methacrylate resins such as bis-GMA, UDMA, and TEGDMA cause the inhibition of DNA and protein synthesis in fibroblast cells and, at higher concentrations, may lead to cell death in the surrounding tissues [9]. In the cited in vitro study, the authors demonstrated that most of adhesive resins used in orthodontics and available on the European market release biologically harmful substances to the immediate environment [10]. Therefore, the clinical verifiability of widely used dental bonding systems including OptiBond Solo Plus is still necessary, as despite their postulated harmful effect on living tissues, these materials are still used in everyday clinical dental practice [2]. The studies on new polymerization and filling strategies for dental composites in a mouse model demonstrated that composite materials are also indicated as possibly being highly cytotoxic, exhibiting detrimental effects on cells [11,12]. These cytotoxic effects include physiological and pathological cellular changes as well as the reactions and proliferation of the cell cultures, observed as pathologically altered cells including fibroblasts with nuclear enlargement, binucleation, nuclear anomalies, and dead cells including rounded fibroblasts, cells with nuclear disintegration, and pyknoses [13–15]. Or MPST, 3-mercaptopyruvate sulfurtransferase and CTH, gamma-cystathionase, belong to the class of enzymes called sulfurtransferases, which transfer sulfur atoms and are involved in $H_2S$/sulfane sulfur endogenous formation from L-cysteine. MPST is a housekeeping and multifunctional enzyme with antioxidative function. It has a role in hydrogen sulfide and polysulfide production, and possibly sulfur oxide production. Other sulfur enzymes involved in the biochemical production of hydrogen sulfide are CBS, cystathionine β-synthase, and TST, rhodanese. It has been described that sulfurtransferase, due to the presence of free-SH groups, shows local antioxidant activity [16,17]. $H_2S$ metabolism involving all of the above enzymes is illustrated in Figure 1.

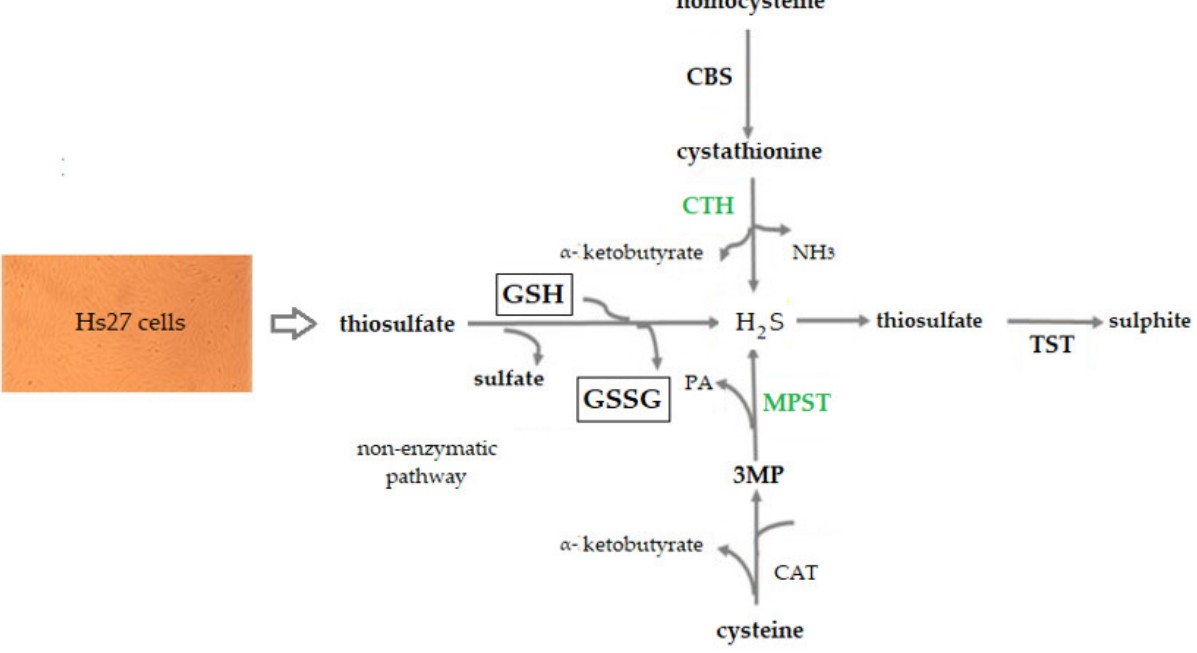

**Figure 1.** $H_2S$ metabolism in Hs27 cell lines. CBS—cystathionine β-lyase, CTH—γ-cystathionase, GSH—glutathione reduced form, GSSG—glutathione oxidized form, MPST—3-mercaptopyruvate sulfurtransferase, TST—rhodanese, CAT—cysteine aminotransferase, 3MP—3-mercaptopyruvate, PA—pyruvate.

The null hypothesis of the present study was that sulfur enzyme expression levels do not change upon OptiBond Solo Plus incubation and provide no insights into the OptiBond Solo Plus induced processes

## 2. Materials and Methods

### 2.1. OptiBond Solo Plus Solution Preparation

A set of experiments was conducted to determine the optimal solvent for the OptiBond Solo Plus (Kerr, Orange, CA, USA) suspension. Water, ethanol, warm and cold culture medium, and DMSO (dimethyl sulfoxide), among others, were tested. Finally, in order to introduce the OptiBond Solo Plus to the culture medium, the appropriate amount of suspension was introduced to the warm (incubated in thermal block at 37 °C for 5 min) DMSO (100%) to create 1% stock solution.The desired amount of prepared OptiBond Solo Plus in DMSO was mixed with the culture cell medium and then introduced to the plate with cells (in concentration of 0.003, 0.01, 0.02, and 0.03%).

### 2.2. Cell Cultures

Experiments were conducted in a SANYO incubator at 37 °C in a humidified atmosphere containing 5% $CO_2$. Normal Hs27 neonatal foreskin cell line (ATCC, Manassas, VA, USA) was grown in a monolayer in Dulbecco's modified Eagle medium (DMEM) supplemented with 10% fetal bovine serum (Biowest) and antibiotics (100 U/mL penicillin and 100 μg/mL streptomycin solution, HyClone). Cells were passaged regularly in a sterile chamber with laminar airflow. The cells were dissociated from plastic culture dishes by washing with pre-warm 0.25% trypsin solution, (HyClone). Pre-warmed culture medium was added to inactivate trypsin. The cell suspension was centrifuged at $300–1000 \times g$ for 6 min, supernatant removed, and the cells in fresh culture medium were counted in the Bürker chamber. Cells were seeded onto 100 mm diameter dishes with 9 mL of culture medium ($1.0–1.5 \times 10^6$ cells/1 mL culture medium) or 6-, 24-, 96- multi-well plates according to the experimental needs, then grown until 90% confluency. Cells were incubated in the absence or presence of the tested concentrations of OptiBond Solo Plus (Kerr, Orange, CA, USA). The experiments to determine the appropriate concentrations were performed and a set of concentrations was selected: 0.003, 0.010, and 0.03%. The highest chosen concentration was one visibly showing the effect on fibroblast cells as evaluated under the microscope. Visible physiological cellular changes in cell cultures were observed including rounded fibroblasts, cell shrinkage, and undetached dead cells were present. The literature search was also involved in the process of the proper dilution determination. The information regarding the in vitro cellular model of OptiBond Solo Plus certain dilutions were taken into consideration [2,15].

Each experiment was accompanied by an additional control where DMSO alone—in a final concentration of 0.1%—was added to the cells. This was due to the OptiBond being introduced first into the DMSO solution and then into the culture medium. The DMSO-treated cells were included in the experimental model to observe and prove that the DMSO itself does not affect the cells in any way that could change the results for OptiBond Solo Plus. All solutions were pre-warmed to 37 °C to meet the temperature of the culture medium.

### 2.3. LDH Cell Membrane Integrity Assay

The colorimetric LDH cytotoxicity assay: The Cytotoxicity Detection Kit (Thermo Fischer, Waltham, MA, USA) was used to assess the cytotoxicity of OptiBond Solo Plus to the medium. The principle of the assay is based on the colorimetric measurement of lactate dehydrogenase (LDH) activity released into the cytoplasm. Under physiological conditions, LDH is not released into the environment. Mechanical rupture of the plasma membrane and subsequent cell death leaks LDH out of the cells into the culture medium. This initiates enzymatic reactions with the test substrate branching to produce the colored product formazan. The increase in LDH activity in the culture medium directly correlates with the

amount of colored product formed during incubation. The cells cultured in 96-well plates were tested after the 24- and 48-h incubations with OptiBond Solo Plus and the cytotoxicity assay was conducted according to the manufacturer's protocol (test controls, clear culture medium, and the experimental control for 0.1% DMSO). Experiments were performed at least three times in six technical repetitions. However, the tested concentrations did not allow us to determine a dose–response curve or the standard $IC_{50}$ or other IC values. Spectrophotometric measurement of the amount of product formed at a wavelength of $\lambda = 492$ nm was carried out in a BIO-TEK pellet reader.

*2.4. The Number of Cells by Crystal Violet Staining*

The crystal violet cell staining method described in [18] was used to evaluate the number of cells. For the crystal violet test, cells were cultured in 96- and 24-well plates for the 24 h- and 48 h- incubations with OptiBond Solo Plus at 37 °C. The medium was removed, cells washed with PBS, and fixed with 200 μL 96% ethanol for 15 min. Next, it was stained with 0.5% water crystal violet, washed with water, air dried, and incubated for 30 min with the elution solution. Cell number was proportional to the intensity of the absorbance signal at $\lambda = 540$ nm. Measurements were made on a BIO-TEK pellet reader. The crystal violet stains the live cells on the cell culture medium by penetrating the cells. In addition to the qualitative analysis of the cells, this assay allows for further quantification of cell numbers through spectrophotometric analysis. The dye absorbed into the cells can be released using a decolorizer solution. The color intensity of the decolorizer solution, added to the wells with stained cells, allows for the number of viable cells to be determined. Experiments were performed at least three times in 3–6 technical repetitions.

*2.5. Gene Expression at the mRNA Level by RT-PCR*

Isolation of RNA: Total RNA was extracted with Tri-Reagent (Lab Empire) following the method described by Chomczyński and Sacchi [19]. Reverse transcription: 3 μg of isolated RNA was reverse transcribed with Reverse Transcriptase in 5× reaction buffer (GoScript<sup>TM</sup> Promega, Promega, Madison, WI, USA) with $MgCl_2$, RNAse Inhibitor and dNTP mix (Thermo Scientific, Waltham, MA, USA) in 10 μL of the final volume of the reaction mixture following the Promega manufacturer's protocol. Polymerase chain reaction (PCR): PCR was performed using a mixture of cDNA, adequate revers (R) and forward (F) primers, DNA polymerase in Tris–HCl pH 8.8 with $MgCl_2$, KCl, Triton X-100, dNTP mix (Thermo Scientific), and $H_2O$-DEPC in a total reaction volume of 12.5 μL. PCR products were analyzed in 2.0% agarose gels during electrophoresis and imaged with UVI-KS 4000 i/ImagePC (Syngen Biotech). The expression of the following enzymes rhodanese (TST), 3-mercaptopyruvate sulfur transferase (MPST), cystathionine gamma-lyase (CTH) and cystathionine beta-synthase (CBS) was checked, and the experiment was conducted according to the protocol described by Bronowicka-Adamska and Wróbel [20]. Beta-actin was used as a reference gene. Experiments were performed at least two times in three technical repetitions The detailed conditions for the PCR run are listed in Table 1.

**Table 1.** PCR conditions.

| Gene | Initiation | Denaturation | Amplification | Elongation | Termination |
|---|---|---|---|---|---|
| MPST | 5 min at 95 °C | 30 s at 95 °C | 30 s at 55 °C | 2 min at 72 °C for 29 cycles | 72 °C for 8 min |
| CTH | 5 min at 95 °C | 30 s at 95 °C | 1 min at 51 °C | 8 min at 72 °C for 30 cycles | 72 °C for 8 min |
| TST | 5 min at 95 °C | 30 s at 95 °C | 30 s at 54.5 °C | 2 min at 72 °C for 34 cycles | 72 °C for 8 min |
| β-actin | 5 min at 95 °C | 30 s at 95 °C | 30 s at 55 °C | 2 min at 72 °C for 30 cycles | 72 °C for 8 min |

*2.6. Gene Expression at the Protein Level by Western Blot*

(a)    SDS-PAGE electrophoresis

Carried out in Laemmli's buffer system on 4% concentration gel and on 10 and 12% separation polyacrylamide gels in the presence of SDS (60 V and 120 V voltage) [21]. The

amount of protein applied in the wells was predicted to be 15 to 30 μg depending on the experiment on BioRad's Mini Protean 4 Cell apparatus.

(b)    Western Blot with immunohistochemical identification of proteins

After electrophoretic separation, the gel was electrotransferred overnight at 4 °C in a semi-wet system to a PVDF (polyvinylidene fluoride) membrane at 0.15 A. After completion of the transfer, the membrane was blocked for two hours at room temperature by incubation in TBS-Tween-20 solution with 5% non-fat milk powder after washing with TBS-Tween-20. After rinsing, incubation with primary antibody anti-MPST, anti-CTH, and others overnight at 4 °C was conducted. Anti-beta actin, and anti-alpha-tubulin antibodies were used to check for loading (Abnova, Taipei, Taiwan; GeneTex, Irvine, California, USA ProteinTech, Manchester, UK; 1:1000). Next, the membrane was washed and incubated with a secondary anti-mouse or anti-rabbit antibody conjugated with alkaline phosphatase (ProteinTech, 1:2000). The final step was the development of proteins on the membrane by incubation in a solution of NBT/BCIP substrate (nitro blue tetrazolium chloride/5-bromo-4-chloro-3-indolyl-phosphate, toluidine salt, Roche) dissolved in the buffer at pH = 9.5 until a precipitated color product appeared, indicating the presence of protein binding to the primary antibody. Experiments were performed at least two times with two WB runs for each experiment.

### 2.7. RP-HPLC (Reverse Phase High Performance Liquid Chromatography)

The level of the reduced (GSH) and oxidized glutathione (GSSG) in the incubation mixtures was determined using the RP-HPLC method of Dominick et al. (2001) [22] with modifications [23].

### 2.8. Data Analysis

Results were expressed as the means ± SD. Each experiment was repeated two to six times with multiple technical repetitions.

## 3. Results

The aim of the presented study was the determination of concentrations and incubation times of the dental bonding system OptiBond Solo Plus in which its non-polymerized form would induce a cytotoxic effect and in which it induces changes in the expression levels of sulfur enzymes, with the emphasis on CTH and MPST. Reduced and oxidized glutathione and the amount of living cells were also evaluated. The cytotoxic effect of a wide range of OptiBond Solo Plus concentrations in culture medium was measured by the LDH cytotoxicity test. Table 2 presents the three tested concentrations of OptiBond Solo Plus: 0.003% (OptiBond Solo Plus in DMSO), 0.010%, and 0.03% as well as the control (0.1% DMSO-treated cells). The higher tested concentrations were so potent they left no living cells in the culture medium to investigate (results not shown). Thus, the highest examined concentration was set to 0.03%. The cytotoxic effect was demonstrated for 0.03% OptiBond Solo Plus after the 24 h- and 48 h-incubation times (Table 2).

**Table 2.** Changes in the integrity of Hs27 fibroblast cell membranes after the 24- and 48-h incubation with OptiBond Solo Plus tested by the LDH Cytotoxicity Kit. Data referenced to values obtained for the control (0.1% DMSO-treated cells).

| OptiBond Solo Plus | % Cytotoxicity | Time |
|:---:|:---:|:---:|
| 0.003% | <5% | |
| 0.010% | <5% | 24 h |
| 0.03% | 43.7 | |
| 0.003% | <5% | |
| 0.010% | <5% | 48 h |
| 0.03% | 41.0 | |

Following this, the crystal violet staining protocol was introduced, and the pictures of cell cultures treated with OptiBond Solo Plus were taken. The reduced number of cells was registered for concentrations above 0.010% after the 24-h incubation, 48-h incubation and for all tested concentrations after the 48-h incubation. At maximum, the amount of living fibroblast cells decreased down to 50% (in comparison to control listed as 100%) after the 48-h incubation with the highest concentration of OptiBond Solo Plus (Figure 2). The significant change in the number of cells was also observed on the microscopic scale, as visualized in the cell culture pictures (not shown).

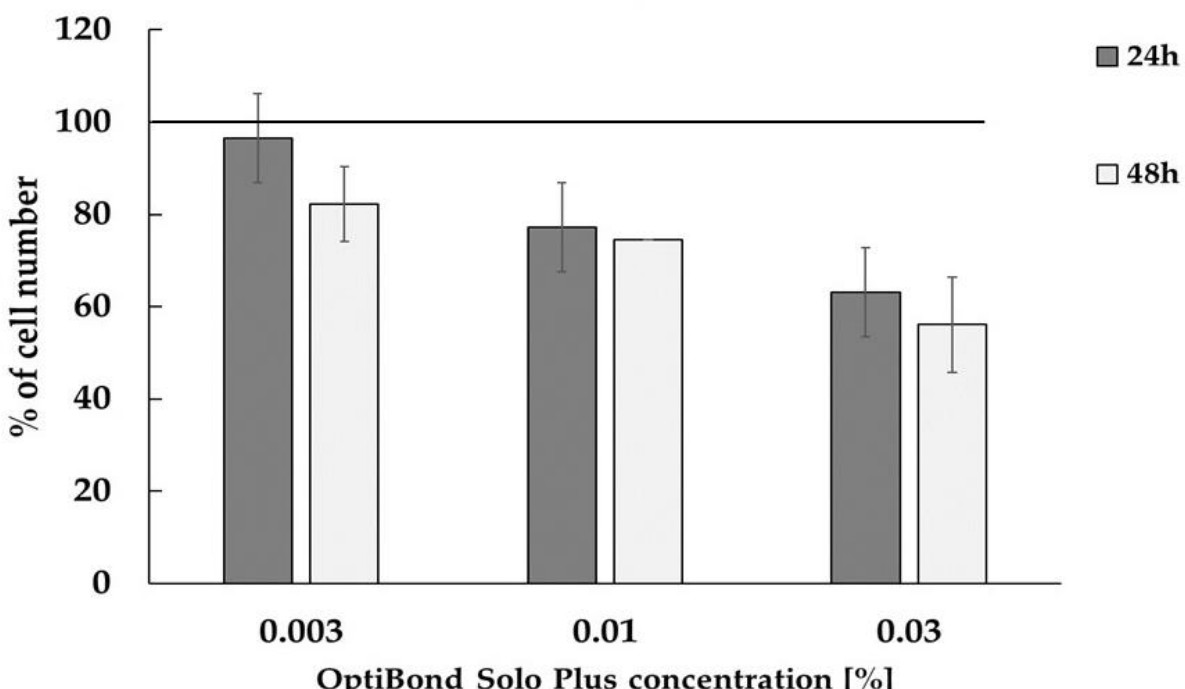

**Figure 2.** Cell number in Hs27 cell line after the 24- and 48-h incubations with OptiBond Solo Plus by crystal violet staining. The line at the 100% level represents the number of cells stained at the time when all OptiBond concentrations were added to the cell culture medium. The data represent a mean with SD values from at least three independent experiments with technical repetitions.

Next, subsequent experiments including the testing of chosen gene expression were conducted. Due to the strong effect of the highest 0.03% concentration on the cell amount, collection of the cells to the following tests was very difficult, and the highest analyzed concentration was lowered to 0.02%. The tests were performed using RT-PCR and Western blot methods. Changes in the expression of the analyzed mRNA and protein levels were visible after the 48-h incubation for MPST (Figure 3c). The expression of this gene increased after incubation with growing concentrations of OptiBond Solo Plus. The MPST changed expression and the anticipated role in anti-oxidative response encouraged the authors to examine the activity of enzymes participating in the metabolism of sulfur containing compounds. Changes in the expression of the analyzed mRNA were registered after 24 and 48 h of incubation for CTH (Figure 3b). The increase was registered after incubation with growing concentrations of OptiBond Solo Plus for the higher concentration. Additionally, there were no conclusive changes in the expression of TST and CBS genes (results not shown) at the mRNA and protein level.

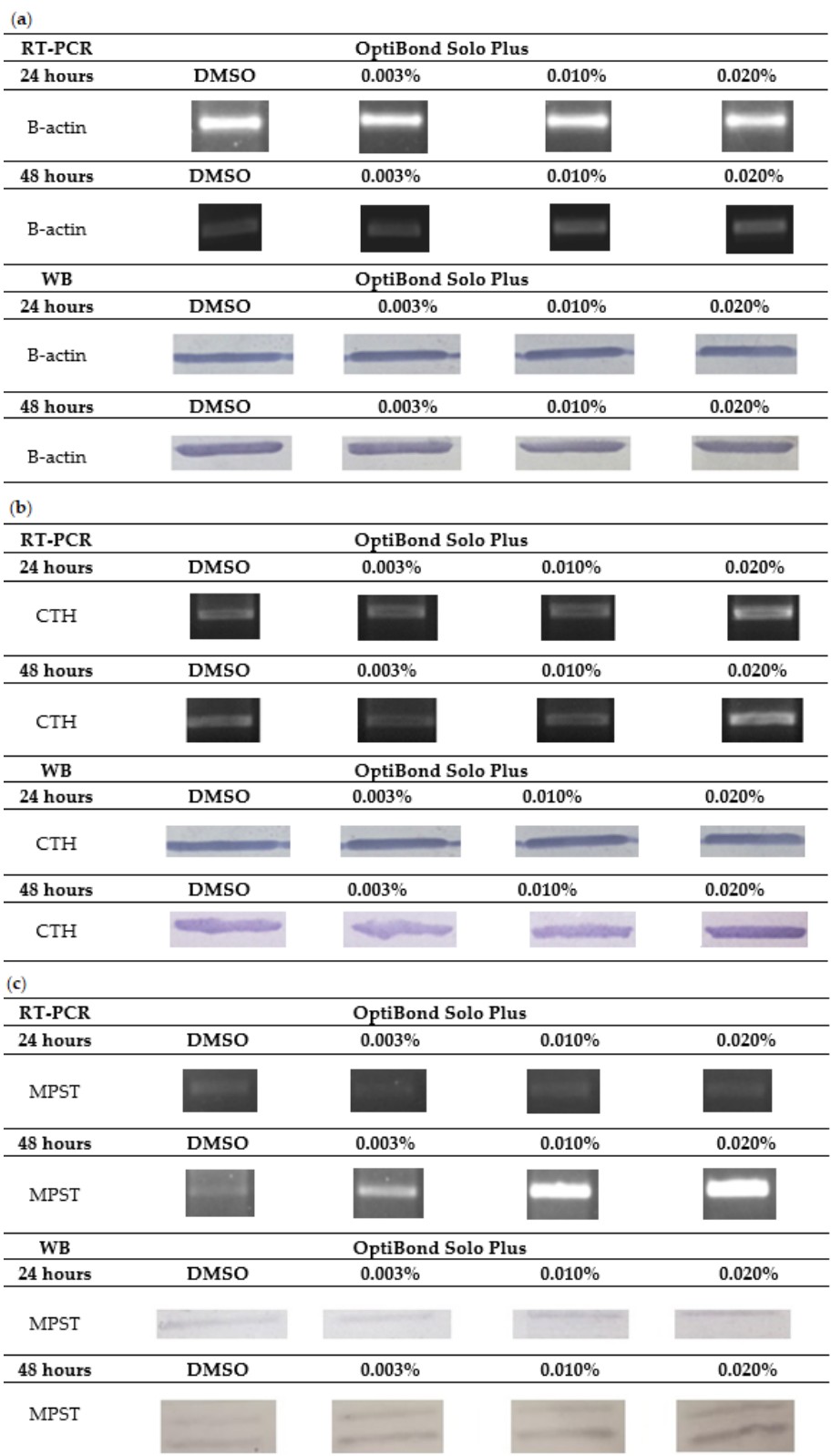

**Figure 3.** Expression of (**a**) β-actin, (**b**) CTH, and (**c**) MPST in Hs27 fibroblast cells after 24- and 48-h incubations with OptiBond Solo Plus by RT-PCR and Western blot (WB). The tested concentrations of OptiBond Solo Plus in culture medium as listed: 0.003, 0.010, and 0.02 with 0.1% DMSO treated cells as a control. The experiment was repeated at least two times and the WB and the RT-PCR was run two times for the samples from each experiment with similar results. A representative result is shown.

For the highest concentration, we introduced the testing of glutathione levels. The levels of reduced glutathione (GSH) and its oxidized form (GSSG) were measured by modified RP-HPLC. Table 3 shows the level of GSH and GSSG in Hs27 fibroblast cells after the incubation with DMSO (control) and 0.03% OptiBond Solo Plus. Total glutathione, calculated as a sum of 2GSSG content + GSH content, increased significantly in Hs27 fibroblast cells after incubation with 0.03% OptiBond Solo Plus. The increased level of the total glutathione in Hs27 fibroblast cells after incubation with 0.03% OptiBond Solo Plus (225.1 nmol g of protein$^{-1}$) in comparison to the control group (DMSO) of 98.1 nmol mg of protein$^{-1}$ confirmed its production in cells. Low GSH/GSSG ratio in Hs27 fibroblast cells after incubation with 0.03% OptiBond Solo Plus showed that the equilibrium moved toward the oxidized form. GSH is the major non-protein thiol in mammalian cells and plays a central role as an antioxidant.

**Table 3.** The levels of reduced (GSH) and oxidized (GSSG) glutathione in small volume samples by modified RP-HPLC. The tested concentration of OptiBond Solo Plus in culture medium—0.03%. Experiments were run three times. The results from the representative experiment are shown. GSH = glutathione reduced, GSSG = glutathione oxidized. The data represent a mean with the SD value.

|  | GSSG | GSH | GSH/GSSG |
|---|---|---|---|
| 0% | 7.24 ± 1.18 | 83.62 ± 3.46 | 11.54 |
| 0.03% | 29.06 ± 6.9 | 166.97 ± 6.6 | 5.74 |

## 4. Discussion

The projects' goal was to study the influence on cytotoxicity, cell number, and expression of selected sulfur enzymes, with particular emphasis on CTH and MPST upon the exposure of OptiBond Solo Plus. The findings, with additional determination of GSH and GSSG levels, were the opening point for a discussion on how the changing OptiBond Solo Plus concentration is related to oxidative stress.

Fibroblast cell line was chosen as an established model for basic research regarding the dental materials. The decision to test normal fibroblast cell line Hs27 was made in accordance with the key role of fibroblast cells in damaged tissue reparation such as the ability to migrate to the damaged part of the skin and proliferate in order to repair the injury [24]. The OptiBond family adhesive systems are one of the most widely used in dental practice [1]. Its effectiveness has been well described in caries inhibition in vitro [24,25] and its usage is being tested in current experiments [2,26]. We decided to investigate the cytotoxic effect and effect on cell number upon incubation with OptiBond Solo Plus, mostly because there is no consensus in the literature regarding the toxicity of dental adhesive systems toward dental pulp to date [2,27]. In our study, the cytotoxic effect was demonstrated for the concentration of 0.03% after both incubation times, while the reduced number of cells was registered in a dose dependent manner, visibly observed in the microscopic picture. The experiments involving cytotoxicity and strength of clinically used dental bonding systems are described in the literature in in vitro models with dentine discs, dentine barrier models, and pulp chamber dentin [28–30]. However, a significant number of papers have described the in vitro model such as ours, where adhesives were tested in the culture cell medium [2,17,24]. This ongoing research and described subject exploration requirements justify our chosen experimental model. The authors decided to evaluate unpolymerized bonding materials as they are applied in this form in close proximity to the dental pulp, then sprayed in order to be pushed into dentinal canaliculi and penetrate even further toward the pulp tissue, prior to polymerization. Therefore, in the closest possible way, it reflects the clinical situation, as—in the first phase of restoration with dental composites—bonding agents are applied in unpolymerized form and can indeed irritate dental pulp tissue when in contact with this cell-rich connective tissue. Additionally, unpolymerized bonding agents express higher cytotoxicity than the ones that were polymerized properly, therefore, it was

crucial to assess the clinical safety of OptiBond in the worst case scenario (e.g., improper polymerization technique). Moreover, polymerized dental products break down over time and release monomers, potentially exposing surrounding tissues including pulp to a noxious effect [2,4,5]. Our results correspond to the findings of those described in the literature. In the paper by Taubmann, human gingival fibroblast—hGF—condition after 24 h of incubation with dilutions of several dental adhesives were tested. A concentration-dependent reduction in cell viability XTT assay upon exposure of dental adhesives including OptiBond Solo Plus was shown [2]. Santos conducted the experiments on 3T3 mouse fibroblast cells and MG-63 osteoblast-like cells (human osteosarcoma cell lines), which were exposed for a 24-h incubation with OptiBond Solo Plus in different dilutions. The authors discussed the differences in the toxicity of three dental adhesives and showed the results of the MTT test regarding the range of concentrations listed as dilutions $v/v$ from 1:50 to 1:1. They observed the significant decrease in the number of living cells for OptiBond Solo Plus dilutions higher than 1:50 (over 0.2%), which is consistent with what we observed in our experiments. However, in our study, the concentrations higher than 0.3% were so potent that they left no living cells in the culture medium to investigate. In the paper by Santos, OptiBond Solo Plus was described as the most toxic toward the 3T3 fibroblast cell line and as such, is still one that needs further investigation [18]. In comparison, OptiBond Universal, but in the polymerized, safer form, was tested in an in vitro cellular experimental model—monocyte/macrophage cell line SC—where the cytotoxicity was measured by means of the XTT assay [25]. Still, the authors concluded that it presented significant cytotoxicity and genotoxicity toward the SC cell line and showed a significant ability to induce apoptosis in the SC cell line. The both mentioned adhesives were described by Pagano as cytotoxic to primary human fibroblast cells derived from patients with a time and concentration dependent cytotoxic effect and the greatest cell damage among the tested dental adhesives. Following the cytotoxicity tests, the same author investigated the expression levels of chosen genes, which they assessed by RT-PCR as well as Western blot analysis [31]. OptiBond was described to increase gene expression of inflammatory markers: IL1β, IL6, and IL8, which corresponded to the increase in the activated form NF-kB [31]. In our previous studies regarding the effect of fluoride ions on the fibroblast cell line Hs27, we confirmed the expression of four enzymes involved in sulfur metabolism (MPST, CTH, CBS, and TST) using both the RT-PCR and Western blot methods [32]. 3-Mercaptopyruvate sulfurtransferase and γ-cystathionase were closely investigated in the paper. Recent research regarding sulfurtransferases including MPST, TST, and CTH point to their role in the responses to oxidative stress. Studies confirmed the anti-oxidative effect of $H_2S$ produced by these enzymes. In addition, it has been described that sulfurtransferases, due to the presence of free-SH groups, may also show a local antioxidant activity. MPST activity is regulated by redox change and increases under reducing conditions [16,17,20]. We suspected that our study on the level of sulfurtransferase expression could be linked to the long-term responses to the cytotoxic effect of OptiBond Solo Plus. Thus, the chosen enzymes involved in the metabolism of sulfur-containing compounds were tested to connect the cytotoxicity of OptiBond Solo Plus concentrations with a possible anti-oxidative response. In the presented paper, the 48-h incubation with the highest OptiBond Solo Plus concentration increased the intensity of the MPST mRNA bar significantly in comparison to the control. The results were confirmed on the protein level. Expression of this gene increased after incubation with growing concentrations of OptiBond Solo Plus. We also registered the change in the expression of the analyzed mRNA for both incubation times with the highest OptiBond concentration for the CTH enzyme. Our previously studied effect of fluoride ions on this set of enzymes showed neither change on the mRNA level of MPST, TST, and CTH nor the change in the protein level upon the incubation with $F^-$. We then concluded that in the case of the Hs27 cell line, the cytotoxic effect was not connected with the changes in sulfurtransferases, whose elevated levels could show the antioxidant response to fluoride ions [32]. Thus, our clear results encouraged us to set a hypothesis

about the significance of the above enzymes in antioxidative response to the examined dental bonding system.

Dental materials including dental resin monomers may take part in the processes of generating reactive oxygen species (ROS) while in contact with the living tissue of teeth. They have the ability to lower the enzymatic activity of antioxidant proteins (such as SOD or catalase) that play a crucial role in coping with free radicals in the main oxidative systems [32–35]. Cytotoxic concentrations of dental adhesives are connected with their ability to cause oxidative stress [36]. Krifka and coauthors suggest the enhancement of the ROS formation, caused by biological reactions activated by dental composites and resin monomers [37]. ROS generation was also described upon the exposure to OptiBond Solo [38].

Oxidative stress leads to a quantitative shift in GSH to GSSG ratios and elevates the cellular redox potential [39]. It has been described that resin monomers cause an intracellular redox glutathione level depletion, and increase in the ROS formation [40]. A consequence of glutathione level decrease is an increased hydrogen peroxide formation. This induces catalase, which causes SOD expression inhibition [35]. Moreover, 0.1 and 0.25 mM of resin monomer BisGMA caused a significant depletion in the intracellular GSH content after 18 h of incubation with the monobromobimane assay [41]. It has also been proven that resin monomers deplete intracellular glutathione stock [40]. In our presented study, we observed total glutathione increase upon the exposure with 0.03% OptiBond Solo Plus. GSH/GSSG ratio in Hs27 fibroblast cells after the incubation with 0.03% OptiBond Solo Plus showed that the equilibrium moved toward the oxidized form. As GSH is the major non-protein thiol in mammalian cells and plays a central role as an antioxidant, we observed that incubation with OptiBond Solo Plus triggered an interesting response in Hs27. The GSH/GSSG ratio decrease in comparison to the control may lead to stating the occurrence of oxidative stress. The observed changes in the total glutathione content may be a result of cellular compensatory mechanisms. Given this, for a certain conclusion, further cysteine level monitoring is required in this experimental model.

So far, published research on the OptiBond Solo Plus agent has focused mainly on compound cytotoxicity tests without considering the broader approach presented in the paper—sulfur enzyme level investigation. It seems that in the case of the human fibroblast Hs27 cell line, the potential cytotoxic effect may be connected with the changes in sulfur enzymes MPST and CTH, whose elevated levels could possibly show an antioxidant response to OptiBond Solo Plus. Our future plans include an investigation of the hydrogen sulfur levels as well as cysteine levels upon OptiBond Solo Plus exposure. More detailed examination of sulfurtransferase expression levels and monitoring of hydrogen sulfide concentrations with additional tests involving catalase or superoxide dismutase SOD may help to further investigate and explain the changes in MPST and CTH expression induced by OptiBond Solo Plus. More tests might confirm the potential association with oxidative stress and allow for conclusions to be drawn regarding cellular inflammation. Ultimately, with further proper in vivo studies, it might allow for clinical conclusions from the aspect of the safety of the bonding agents' administration in everyday dental practice.

## 5. Limitations

We are aware that there are existing limitations to this study. The in vitro cellular model cannot be transferred directly into clinical model experiments and cannot be taken as a proven result for dental practice. This particularly concerns the differences in the in vitro model concentrations in comparison to the in vivo conditions. For our future studies, a broad panel of tested concentrations, allowing us to create a dose-dependent standard curve and $IC_{50}$ value, will be desired as well as a methodology change that will allow for the direct comparison of the concentration effect through all of the tested methods. Thus, both further in vitro and in vivo tests are required for assumptions to be made regarding the clinical approach in dental practice.

## 6. Conclusions

Within the limitations of this study, we concluded that our studies provide new information on the increase in MPST and CTH enzyme expression upon the incubation with OptiBond Solo Plus monomers. With more tests, we may confirm the potential association with oxidative stress and allow for conclusions to be drawn regarding cellular inflammation. Ultimately, with further proper in vivo studies, it might allow for clinical conclusions in the aspect of the safety of the bonding agents' administration in everyday dental practice.

**Author Contributions:** Conceptualization—M.W., K.K. and J.Z.; Methodology—A.B.-I., K.K. and P.B.-A.; Writing-original draft preparation—A.B.-I. and P.B.-A.; Writing-review and editing—M.W. and J.Z.; Editing—K.K. and B.C.; English proofreading—K.K. and B.C.; Visualization—A.B.-I. and P.B.-A.; Supervision—M.W.; Funding acquisition—A.B.-I. All authors have read and agreed to the published version of the manuscript.

**Funding:** This work was financially supported by the Polish Ministry of Science and Higher Education (funder), grant K/DSC/003573 (funding number), of the Jagiellonian University Medical College for young researchers.

**Institutional Review Board Statement:** Not applicable.

**Informed Consent Statement:** Not applicable.

**Data Availability Statement:** Not applicable.

**Acknowledgments:** We would like to thank Dominika Szlęzak (D.S.) and Agnieszka Krawczyk (A.K.) for the technical support in determining the optimal solvent and conditions for the OptiBond Solo Plus suspension and D.S. for work regarding the perfection of the image quality for the paper.

**Conflicts of Interest:** The authors declare no conflict of interest. The funders had no role in the design of the study; in the collection, analyses, or interpretation of data; in the writing of the manuscript, or in the decision to publish the results.

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
