# Peer review of "The Cytotoxicity of OptiBond Solo Plus and Its Effect on Sulfur Enzymes Expression in Human Fibroblast Cell Line Hs27"

_coatings, doi:10.3390/coatings12030382_

Round 1
Reviewer 1 Report
-Title: The title is very long, please modify
-Abstract: a single paragraph without subheading
-Line 36: please correct to [1-3]
-Line 37: you can use the following references to clarify the degradation and the loss of bond strength :Hardan, L.; Bourgi, R.; Kharouf, N.; Mancino, D.; Zarow, M.; Jakubowicz, N.; Haikel, Y.; Cuevas-Suárez, C.E. Bond Strength of Universal Adhesives to Dentin: A Systematic Review and Meta-Analysis. Polymers 2021, 13, 814. https://doi.org/10.3390/polym13050814 Kharouf, N.; Eid, A.; Hardan, L.; Bourgi, R.; Arntz, Y.; Jmal, H.; Foschi, F.; Sauro, S.; Ball, V.; Haikel, Y.; Mancino, D. Antibacterial and Bonding Properties of Universal Adhesive Dental Polymers Doped with Pyrogallol. Polymers 2021, 13, 1538. https://doi.org/10.3390/polym13101538
-Line 49: please add the company, city, country
-Line 51: please correct to [2,7]
-All the references in the text should be prepared following MDPI Style -Line 63: please delete the second point.
- Please add a null hypothésis to the end of intro part
- -Line 81: what is DMSO, please explain
- -There is a lot of spaces into the text, please correct
- -Figure 1 and 2: not clear, and low resolution Figure 3: very low resolution Figure 4 is a table with an image??
- please modify What are the limitations of the present study?
- Please add the conclusion part
- References: Please modify following MDPI Style
Reviewer 2 Report
The authors present an in vitro study evaluating the cytotoxic effects of OptiBond Solo Plus on human fibroblasts. Several techniques were used to characterize the cytotoxic effects.
I suggest the manuscript title be shortened. For instance, removing “study on the” and “dental bonding system”.
The English language needs correction through the text. Several sentences are hard to understand and wordy. This difficult the readers' comprehension. The text should be improved for clarity, with short and direct sentences.
Abstract: please refer to the words in full before using the abbreviations and explain the abbreviations´ meaning.
Lines 26-39: the authors refer to the degradation of the adhesive layer as a source of monomers that can cause cytotoxic. However, incomplete polymerization of the dental adhesives can also be a source of these monomers. Please refer to this in the manuscript.
Several articles have been published regarding the cytotoxic effects of dental adhesives. The introduction should summarize the main findings in this field. The authors refer to cytotoxic effects but should detail them.
OptiBond Solo Plus is a self-etch adhesive. This is never referred to in the manuscript. Several papers compare the adhesives based on their application modes. This should be discussed in the manuscript.
Scheme 1: please change to figure 1 and update the figures numbers. Also, the figure should be referred to in the text.
Lines 69-74: I recommend this information to be removed. The study aim is stated in lines 67-69. The information in lines 69-74 is what was done (materials and methods) and the discussion.
Why did the authors decide to evaluate the unpolymerized adhesive? This does not reflect the clinical conditions, and we can assume the observed cytotoxicity is higher than in the clinical conditions. Please discuss this topic in the discussion section.
Lines 78-79: was OptiBond not soluble in cell culture medium?
Line 81: what was the percentage of DMSO?
Lines 81-82: what do the authors mean by “DMSO was previously incubated for 5 min in 37ºC”?
Lines 84: the authors present volume to volume dilutions here, but later in the results, they refer to percentages. Please uniformize.
It is not clear what the extraction conditions are. Please detail the time, temperature, and other relevant conditions of the extraction.
Line 91: please indicate the cells concentration for each experiment to allow the reproducibility of your work.
Lines 95-96: the authors stated that: “The highest chosen concentration was one visibly showing cytotoxic effect on fibroblast cells”. How was this evaluated?
Did the authors consider testing more concentrations and determining dose-response curves? This had allowed establishing precise cytotoxicity values, for instance the IC50.
Section 2.3: only the 24 hours evaluation time is referred, but table 2 presents the results for the 24 and the 48he evaluations. Please correct it.
Sections 2.3, 2.4, 2.5, 2.6: please complete the information in each subsection. Detail the controls, evaluation times, and other relevant information for each one.
Section 2.4: the crystal violet stains DNA. We can assume this is an indirect viability marker, but direct viability assays exist that were not performed. I suggest changing this subsection title.
Section 2.7: when authors refer to a method with modifications, please refer to the changes performed.
Section 2.7: the authors cite reference 16 as Dominik et al., 2001; however, this reference is not from this author. Please correct it.
The results section needs to be reorganized. The tables and figures should be placed near the corresponding text.
Did the authors perform a statistical analysis of the results? If so, why is this not described? If not, why was it not performed?
Line 165: what control are the authors referring to?
Line 171: if the authors needed to change the higher concentration to 0.02%, why were all the experiments not performed with this concentration instead of 0.03%?
Table 2: the authors refer to 0.1% DMSO, but in section 81 they refer to 1%. This is confusing. Please clarify.
Lines 191-192: please move this information to the discussion section.
Table 2: how do the authors explain that 0.003% presents cytotoxicity and 0.010% does not at 24 and 48 hours? This needs to be addressed in the discussion.
Table 2: how many experiments were performed? What is the SE or SD?
Figure 1: what is being presented? Mean or median? SD or SE? Are the differences statistically significant?
Figure 2: better quality images need to be provided.
Figure 2: the 48 hours images are missing. Please add it.
Figure 3: some images are unreadable. Please replace it with better quality ones.
Figure 3: why did the authors not perform a quantitative evaluation?
Figure 4: the graph should be removed since it presents repeated information from the table.
Figure 4: what is being presented with +/-? SD? SE?
The discussion section needs to be rewritten. It presents several known information but lacks an appropriate discussion of the study results, as, for instance, a possible explanation for the obtained results. Also, add a paragraph referring to the study limitations.
Overall, I think the manuscript needs to be significantly improved. The methods and results need to be reorganized, and the missing information added. In addition, statistical analysis needs to be performed, and images quality improved.
Reviewer 3 Report
- "the appropriate amount of suspension was 80 introduced to the DMSO solution to create 1% stock solution." How can you control the exact amount of the solution in order to have the 1% ?
- "The desired amount of prepared stock solution OptiBond Solo 82 Plus in DMSO solution was mixed with the culture cell medium and then introduced to 83 the plate with cells (volume:volume dilutions of 1:300, 1:100; 1:50 and 1:30). " Why you chose these amounts? Base on what?
- "Cells cultured on dishes were stained with a solution of 0.5% crys- 113 tal violet in deionized water for 2 minutes." This is a protocol? Based on what?
- "Measurements were made on a BIO-TEK python 120 reader, after 24- and 48-hour incubation with OptiBond Solo Plus in medium. " Why you did not evaluate on 12 h, 36 h and 60h?
Round 2
Reviewer 1 Report
The authors modified the paper according to the reviewer comments
Author Response
The Authors would like to thank the Reviewer for accepting the Manuscript.
The spell check of English version was performed and the corrections were made.
Reviewer 2 Report
The authors present an in vitro study evaluating the cytotoxic effects of OptiBond Solo Plus on human fibroblasts. Several techniques were used to characterize the cytotoxic effects. The manuscript was improved from the last revision round; however, improvements and corrections are still needed.
The English language needs correction through the text. Several sentences are hard to understand and wordy. This difficult the readers' comprehension. The text should be improved for clarity, with short and direct sentences. Unnecessary words should be removed. See, for instance, lines 16-17, 20, 38, 128-134, 395-402.
The introduction section needs to be reorganized for logic and clarity. The authors refer to the same topic in different phrases, making this section confusing.
Lines 89-93: I find the study hypothesis long and confusing. I suggest lines 91-93 be removed, and the study hypothesis be simplified.
When asked in the previous revision round about why the authors decided to evaluate the unpolymerized adhesive since it does not reflect the clinical situation, the authors stated that this was added to the study limitations. I do not find the study limitations to reflect this. Furthermore, no reference is made to the use of unpolymerized adhesive and its implications in the study results. The authors should discuss this and why other studies use polymerized adhesives.
Please add the concentration of DMSO to the manuscript. Concentration is not purity; please check that.
Lines 107-117: I find this description unnecessary detailed, and long. I understand a brief description of the cell culture can be helpful, but I find this too long.
Line 118: please refer to cell concentration and not cell number. This would help others to replicate your experimental set.
Lines 124-126: I do not understand this sentence. Please clarify it.
Lines 127-128: please move this sentence to the discussion section.
When asked in the previous revision round about how “The highest chosen concentration was one visibly showing cytotoxic effect on fibroblast cells” was evaluated, the authors stated that “cells were evaluated under the microscope”. Please add this information to the manuscript and detail what was observed to conclude about the cytotoxic effect.
Please add to the study limitations that the tested concentrations did not allow to determine a dose-response curve or the standard IC50 or other IC values.
Section 2.4: why were different plate types used for the same test?
Section 2.4: again, the crystal violet stains DNA, and it’s not a direct viability test. Please correct the section title.
The lack of statistical analysis is a major problem of this manuscript. Therefore, I suggest the authors find a second opinion since I think it is possible to perform a statistical analysis of the obtained results.
Some experiments were performed with 0.03%, and others with 0.02% need to be added to the study limitations since it affects the comparison of the results.
Table 2: the reference to 0.1% DMSO is confusing, since the table refers to 0.003%, 0.01% and 0.03%. When the authors refer to Optibond Solo Plus, it is dissolved in DMSO, right? Please clarify the 0.1% DMSO value used as control.
Table 2: how do the authors explain that 0.003% presents cytotoxicity and 0.010% (which is higher) does not at 24 and 48 hours? This needs to be addressed in the discussion.
Figure and table 4: please indicate in each figure caption what is being represented. Mean or median, SD or SE.
Figure 2: is the data presented normalized to the control value defined as 100%?
Figure 3: figures presents A,B,C and figure caption a,b,c. Please uniformize.
Line 259: I believe the authors meant table 4 instead of figure 4.
Line 282: since the authors refer to the ability of fibroblasts to migrate as important, why was the cells’ migration capacity not evaluated?
The discussion section needs to be rewritten. It presents several known information but lacks an appropriate discussion of the study results, as, for instance, a possible explanation for the obtained results.
The limitations section needs to be incremented. Again, please check the comments above.
Author Response
Point to point responses to the Reviewer 3
Coatings The cytotoxicity of OptiBond Solo Plus and its effect on sulfur enzymes expression in human fibroblast cell line Hs27.
- The English language needs correction through the text. Several sentences are hard to understand and wordy. This difficult the readers' comprehension. The text should be improved for clarity, with short and direct sentences. Unnecessary words should be removed. See, for instance, lines 16-17, 20, 38, 128-134, 395-402.
As for examples:
lines 16-17, 20, changes were introduced according to the Reviewer’s opinion
line 38 was specifically added by other Reviewers request and was accepted as the Manuscript itself in Round 2 revisions so it must remain as it is;
lines 128-134 were added at this Reviewers request to introduce the information and clarify the concept of solvent control
line 395- 402 changes were introduced according to the Reviewer opinion
Our paper was written by authors with the C2 certified level in English language. Our co-author is bilingual, lived, attended school, and studied in the US, where she graduated in Biology with the master’s degree and worked for several years in in the field before moving to Poland. Despite that we sent the text to an experienced English Professor that teaches specifically the writing skills. He introduced minor changes and provided us with their good opinion about the text. We stated in the first round of responses that the verification and corrections were implemented.
Moreover, the other Reviewer marked the English language as acceptable with minor spell check required.
- The introduction section needs to be reorganized for logic and clarity. The authors refer to the same topic in different phrases, making this section confusing.
Thank you for this remark, the introductioion was reorganized accordingly to the Reviewer’s suggetions with no changes to the content as it was introduced accordingly to the other Reviewer‘s demand and was accepted in the Round 2 responces.
- Lines 89-93: I find the study hypothesis long and confusing. I suggest lines 91-93 be removed, and the study hypothesis be simplified.
Thank you for this new remark, the hypothesis was introduced accordingly to the other Reviewers sugesstion and was accepted in the Round 2 responces. Newertheless, we have introduced the suggested elimination of lines 91 to 93.
- When asked in the previous revision round about why the authors decided to evaluate the unpolymerized adhesive since it does not reflect the clinical situation, the authors stated that this was added to the study limitations. I do not find the study limitations to reflect this. Furthermore, no reference is made to the use of unpolymerized adhesive and its implications in the study results. The authors should discuss this and why other studies use polymerized adhesives.
Thank you for this remark. The authors decided to evaluate unpolymerized bonding material, as it is applied in this form in close proximity to the dental pulp, then sprayed in order to be pushed into dentinal canaliculi and penetrate even further towards pulp tissue, prior to polymerization. Therefore, it does reflect clinical situation, as - in the first phase of restoration with dental composites - bonding agents are applied in unpolymerized form and can indeed irritate dental pulp tissue while come in contact with this cell-rich connective tissue. Also, non-polymerized bonding agents express higher cytotoxicity than the ones that were polymerized properly, therefore it was crucial to assess clinical safety of OptiBond in the worst case scenario (e.g., improper polymerization technique). The third reason was that there are solid scientific proofs, indicated in this manuscript, that polymerized dental products break down over time and release un-polymerized components, exposing surrounding tissues - especially the pulp – to a potential noxious effect.
The reference to the use of unpolymerized adhesive and its implications in the study results was added to the manuscript.
- Please add the concentration of DMSO to the manuscript. Concentration is not purity; please check that.
Thank you for this remark. The information was added to the manuscript according to the Reviewer’s suggestion. As a solvent used for preparing OptiBond solution, DMSO was considered to have 100%.
- Lines 107-117: I find this description unnecessary detailed, and long. I understand a brief description of the cell culture can be helpful, but I find this too long.
Thank you for this new remark. The changes were introduced to the text to make the description clear and brief according to the present comment.
This extended description was added, due to this Reviewer’s previous comments about information needed to be added: “Please complete the information in each subsection. Detail the controls, evaluation times, and other relevant information for each one.” “It is not clear what the extraction conditions are. Please detail the time, temperature, and other relevant conditions of the extraction.” But we accept the Reviewer’s opinion that it was not done properly.
- Line 118: please refer to cell concentration and not cell number. This would help others to replicate your experimental set.
Thank you for this remark. The cells are adherent, specific in the shape and size and within all experiments we count them and perform the experiments according to the cell number. This way of description is also consistent with the Authors other publications. We added additional information about confluence of the cell culture and amount of medium used. This will also allow others to replicate our experiments or experimental setup.
- Lines 124-126: I do not understand this sentence. Please clarify it.
Thank you for this remark. The change was made according to the Reviewer’s suggestion.
- Lines 127-128: please move this sentence to the discussion section.
Thank you for this new remark. The change was made according to the Reviewer’s suggestion.
- When asked in the previous revision round about how “The highest chosen concentration was one visibly showing cytotoxic effect on fibroblast cells” was evaluated, the authors stated that “cells were evaluated under the microscope”. Please add this information to the manuscript and detail what was observed to conclude about the cytotoxic effect.
Thank you for this remark. The information was added to section 2.2. of the manuscript according to the Reviewer’s suggestion.
- Please add to the study limitations that the tested concentrations did not allow to determine a dose-response curve or the standard IC50 or other IC values.
Thank you for this remark. The information was added to the section 2.3 of the methodology and in the study limitations according to the Reviewer’s suggestion.
- Section 2.4: why were different plate types used for the same test?
Thank you for this remark. The 96-well plates were used while the cytotoxicity method was performed (the method requires only culture medium for the cytotoxicity tests to be performed) so the cells on the plates were simultaneously tested via crystal violet staining. To check the method replicability the tests were also performed in 24-well plates.
- Section 2.4: again, the crystal violet stains DNA, and it’s not a direct viability test. Please correct the section title.
Thank you again for this remark. The change in the section title as well as within the manuscript was made according to the Reviewer’s suggestion.
- The lack of statistical analysis is a major problem of this manuscript. Therefore, I suggest the authors find a second opinion since I think it is possible to perform a statistical analysis of the obtained results.
Thank you for this remark. But the Authors will not add the statistical analysis to the paper.
The usually used U-Mann Whitney test that fits all the criteria for our small cell culture experimental models was not applicable here due to the lack of enough separate experiment repetitions (from minimum of 9). We know that some researcher’s equal technical repetitions and experiment repetitions, but we don’t want to bend the results to meet the expectations of the presence of statistics in the work. As for the number of experiments for the method used, they are within the good scientific practice, but no time and founds were available to triple the number of experiments.
- Some experiments were performed with 0.03%, and others with 0.02% need to be added to the study limitations since it affects the comparison of the results.
Thank you for this new remark. The information was added to the study limitations according to the Reviewer’s suggestion.
- Table 2: the reference to 0.1% DMSO is confusing, since the table refers to 0.003%, 0.01% and 0.03%. When the authors refer to OptiBond Solo Plus, it is dissolved in DMSO, right? Please clarify the 0.1% DMSO value used as control.
Thank you for this remark. The information was changed into “Data referenced to values ​​obtained for the control (0.1%DMSO-treated cells) “to clarify the control according to the Reviewer’s suggestion.
Yes, it is dissolved in DMSO. In Section 2.2 of Methodology about cell culture the paragraph is currently present (and was added to the previous version of manuscript clarifying the controls used in the experiments) - “DMSO alone - in final concentration of 0.1 % - was added to the cells. It was due to the OptiBond being introduced first into the DMSO solution and then into the culture medium. This DMSO-treated cells were included into experimental model to see and prove that the DMSO itself does not affect the cells in any way that could change the results for OptiBond Solo Plus”.
- Table 2: how do the authors explain that 0.003% presents cytotoxicity and 0.010% (which is higher) does not at 24 and 48 hours? This needs to be addressed in the discussion.
Thank you for this remark. The information in Table 2 was changed, to introduce the information about the lack of cytotoxicity which is true for values lower than 5% and the only aspect of importance in the conclusion. It is also already addressed in the discussion.
- Figure and table 4: please indicate in each figure caption what is being represented. Mean or median, SD or SE.
Thank you for this remark. The information was added according to the Reviewer’s suggestion to Table no 4. Figure no 4 has been removed from this version of manuscript.
- Figure 2: is the data presented normalized to the control value defined as 100%?
Thank you for this remark. The appropriate explanation was added to the description to describe the control.
Yes, indeed it is, and there was a lack of that information in the text. The 100 [%] line present on the Figure 2 represents the number of cells stained at the time when all OptiBond concentrations/DMSO were added to the cell culture medium. We seed the cells on two identical plates, and one is used as a base for initial number of cells present in the time when OptiBond is added.
- Figure 3: figures present A,B,C and figure caption a,b,c. Please uniformize.
Thank you for this remark. The change was added according to the Reviewer’s suggestion and the Figure 3 was modified.
- Line 259: I believe the authors meant table 4 instead of figure 4.
Thank you for this remark. The change was added according to the Reviewer’s suggestion.
- Line 282: since the authors refer to the ability of fibroblasts to migrate as important, why was the cells’ migration capacity not evaluated?
Thank you for this new remark. It is not the focus of our study, our focus was the sulfur enzymes changes, and the migration is not the topic that would constitute an additional value to our study. Ability of fibroblasts to migrate is important regarding the organism and the tissue characteristics and its presence in the oral cavity.
We kindly take this suggestion in consideration for possible future experimental models of the subject.
- The discussion section needs to be rewritten. It presents several known information but lacks an appropriate discussion of the study results, as, for instance, a possible explanation for the obtained results.
With the agreement of all Authors and to their best knowledge and judgment the Discussion is written in an acceptable way. In particular it emphasizes the main finding of the paper regarding the existing change in the sulfur enzyme MPST and CTH expression upon the incubation with the OptiBond Solo Plus and give the possible explanation of the results. It is also put in the context of similar or corresponding data and research described in literature, which justify the chosen methods and undertaken approach.
Despite that, the changes were introduced to this version of manuscript to meet the Reviewer’s demands.
- The limitations section needs to be incremented. Again, please check the comments above.
Thank you for this remark. The changes were added according to the Reviewer’s suggestion.

Round 3
Reviewer 2 Report
The authors present an in vitro study evaluating the cytotoxic effects of OptiBond Solo Plus on human fibroblasts. Several techniques were used to characterize the cytotoxic effects. The manuscript was improved from the last revision round; however, improvements and corrections are still needed.
The introduction section needs to be reorganized for logic and clarity. The authors refer to the same topic in different phrases, making this section confusing.
Lines 81-85: I find the study hypothesis long and confusing. The authors refer to “would induce a cytotoxic effect”, so these are results and not the study aim.
Line 118: please refer to cell concentration (in cells per ml) and not cell number. This would help others to replicate your experimental set.
The lack of statistical analysis is a major problem of this manuscript. Therefore, I suggest the authors find a second opinion since I think it is possible to perform a statistical analysis of the obtained results.
Table 2: the authors decided to remove the 0% values from the 0,010% concentration, claiming values below 5% correspond to the lack of cytotoxicity. Although I agree with this, still it’s hard to understand why higher concentration values provide no cytotoxic effects regarding lower ones.
Author Response
Coatings The cytotoxicity of OptiBond Solo Plus and its effect on sulfur enzymes expression in human fibroblast cell line Hs27.
The introduction section needs to be reorganized for logic and clarity. The authors refer to the same topic in different phrases, making this section confusing.
Thank you for this remark. Small changes were introduced to the manuscript according to the Reviewers suggestion.
Lines 81-85: I find the study hypothesis long and confusing. The authors refer to “would induce a cytotoxic effect”, so these are results and not the study aim.
Thank you for this remark. The mentioned fragment was moved to the Results section according to the remark
Line 118: please refer to cell concentration (in cells per ml) and not cell number. This would help others to replicate your experimental set.
Thank you for this remark. The information was added to the manuscript according to the Reviewers suggestion.
The lack of statistical analysis is a major problem of this manuscript. Therefore, I suggest the authors find a second opinion since I think it is possible to perform a statistical analysis of the obtained results.
We are also firm about our last answer (included below) about the lack of statistical analysis.
“The usually used U-Mann Whitney test that fits all the criteria for our small cell culture experimental models was not applicable here due to the lack of enough separate experiment repetitions (from minimum of 9). We know that some researcher’s equal technical repetitions and experiment repetitions, but we don’t want to bend the results to meet the expectations of the presence of statistics in the work. As for the number of experiments for the method used, they are within the good scientific practice, but no time and founds were available to triple the number of experiments.”
Table 2: the authors decided to remove the 0% values from the 0,010% concentration, claiming values below 5% correspond to the lack of cytotoxicity. Although I agree with this, still it’s hard to understand why higher concentration values provide no cytotoxic effects regarding lower ones.
Thank you for this remark.But the main point of this study requires only the statment about the lack or the presence of cytotoxicity.